# Ultrafast isomerization-induced cooperative motions to higher molecular orientation in smectic liquid-crystalline azobenzene molecules

Masaki Hada [1,2,6], Daisuke Yamaguchi [3,6], Tadahiko Ishikawa [4], Takayoshi Sawa[2], Kenji Tsuruta [2], Ken Ishikawa [5], Shin-ya Koshihara[4], Yasuhiko Hayashi[2] & Takashi Kato [3]

The photoisomerization of molecules is widely used to control the structure of soft matter in both natural and synthetic systems. However, the structural dynamics of the molecules during isomerization and their subsequent response are difficult to elucidate due to their complex and ultrafast nature. Herein, we describe the ultrafast formation of higher-orientation of liquid-crystalline (LC) azobenzene molecules via linearly polarized ultraviolet light (UV) using ultrafast time-resolved electron diffraction. The ultrafast orientation is caused by the *trans*-to-*cis* isomerization of the azobenzene molecules. Our observations are consistent with simplified molecular dynamics calculations that revealed that the molecules are aligned with the laser polarization axis by their cooperative motion after photo-isomerization. This insight advances the fundamental chemistry of photoresponsive molecules in soft matter as well as their ultrafast photomechanical applications.

[1] Tsukuba Research Center for Interdisciplinary Materials Science (TREMS), Faculty of Pure and Applied Sciences, University of Tsukuba, Tennodai 1-1-1, Tsukuba 305-8573, Japan. [2] Graduate School of Natural Science and Technology, Okayama University, Tsushima-naka 3-1-1, Kita-ku, Okayama 700-8530, Japan. [3] Department of Chemistry & Biotechnology, School of Engineering, The University of Tokyo, 7-3-1, Hongo, Bunkyo-ku, Tokyo 113-8656, Japan. [4] School of Science, Tokyo Institute of Technology, 2-12-1 Ookayama, Meguro-ku, Tokyo 152-8551, Japan. [5] School of Materials and Chemical Technology, Tokyo Institute of Technology, 2-12-1 Ookayama, Meguro-ku, Tokyo 152-8550, Japan. [6] These authors contributed equally: Masaki Hada, Daisuke Yamaguchi. Correspondence and requests for materials should be addressed to M.H. (email: hada.masaki.fm@u.tsukuba.ac.jp) or to T.K. (email: kato@chiral.t.u-tokyo.ac.jp)

The isomerization of photochromic molecules can induce dynamic structural changes in soft matter. The isomerization of retinal in the photoreceptor protein rhodopsin, which enables vision, is an example of this type of system in nature[1]. In synthetic systems, the drastic structural changes induced by isomerization are used to control the structures of soft materials, such as liquid crystals[2], micelles[3,4], vesicles[5,6], thin films[7,8] and surfactant systems[9], via order-to-disorder transitions. The photoisomerization of a single molecule occurs on the timescale of hundreds of femtoseconds[10–15], and the subsequent response of soft matter may initiate in a few picoseconds. However, to the best of our knowledge, neither the ultrafast structural dynamics of the photoisomerization of molecules in the liquid-crystalline (LC) phase nor the subsequent responses have been directly observed. Here, we report the ultrafast orientation in a single direction of LC azobenzene molecules in a smectic B phase using picosecond time-resolved electron diffraction and model calculations based on molecular dynamics.

Liquid crystals combine the fluidity of liquids and the order of solid crystals[16]. The ordered states of liquid crystals can be controlled by photostimuli if photochromic molecules are incorporated into the material[17–25]. Azobenzene molecules are representative photochromic molecules that change from straight *trans*-forms to bent *cis*-forms via photoisomerization[26–28]. LC molecular assemblies containing azobenzene molecules have various applications, e.g., photomechanical materials[17,18], photoswitching devices[19–21], storage devices[22] and molecular photoalignment processes[23–25].

Ultrafast time-resolved diffraction provides structural information about the atomic rearrangements during the phase transitions of materials or chemical reactions between molecules, making it an ideal methodology for investigating photoinduced collective molecular motions[29,30]. Great advances in ultrashort-pulsed and ultrabright electron sources have been made recently, and time-resolved electron diffraction can now be used to elucidate the structural dynamics of organic molecular crystals[31]. Very recently, this technique has been applied to more complex soft materials[32]. The difference between the diffraction patterns before and after photoexcitation reveals the molecular-level structural change of interest in soft matter.

In the present study, we examine the molecular motion of photoresponsive LC azobenzene molecules by picosecond time-resolved electron diffraction measurements and molecular dynamics calculations. Our findings show that the *trans*-to-*cis* isomerization induced in the azobenzene-molecule-based LC assembly via linearly polarized ultraviolet (UV) light induces ultrafast higher orientation.

## Results

**System design and its behaviour**. The photoresponsive molecular system used in this study is shown in Fig. 1. The loosely packed azobenzene molecules **1** are irradiated with a pulse of linearly polarized UV light. A fraction of the homeotropically aligned LC azobenzene molecules in the smectic B phase undergo *trans*-to-*cis* photoisomerization. In the LC phase, the transition dipole moments of azobenzene discs (from the top view of Fig. 1) are randomly oriented; the photoisomerization of azobenzene molecules occurs in the specific direction resulting from the selective excitation of the azobenzene molecules whose transition dipole moments are aligned with the laser polarization axis. This selective isomerization and the subsequent steric effects in one direction induce molecular alignment (higher orientation). The flexibility of the liquid crystal makes this phenomenon efficient: the motion occurs on the timescale of 100 ps, which is the fastest intermolecular motion ever observed in soft matter. Previously,

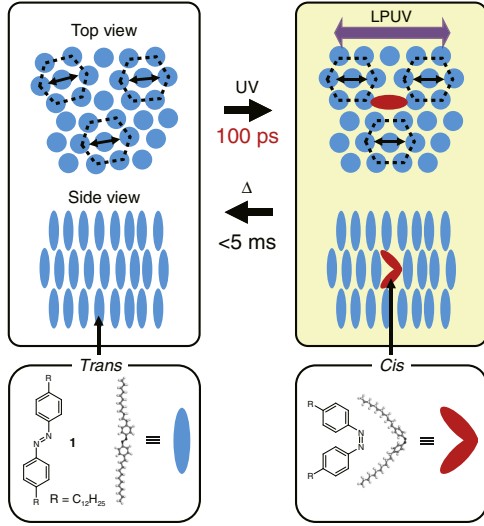

**Fig. 1** Schematic illustration of ultrafast molecular orientation in an LC assembly. The azobenzene molecules (spheroids) in the LC assembly are aligned with the laser polarization axis on the timescale of 100 ps. The reverse reaction occurs within 5 ms via thermal stabilization. This phenomenon is initiated by the *trans*-to-*cis* isomerization of an azobenzene molecule **1**. The red bent spheroids represent azobenzene molecules in the *cis*-form. LPUV is linearly polarized UV light, and the direction of the angle is the polarization axis

photoisomerization behaviours have always been shown to induce slow order-to-disorder transitions in LC assemblies containing azobenzene molecules; however, our system is designed to undergo a new cooperative motion. We chose an azobenzene molecule, 4,4′-didodecylazobenzene (Supplementary Note 1), with an azobenzene core and flexible alkyl chains to prepare the LC phase[33]. The differential scanning calorimetry curves (Supplementary Fig. 1), X-ray diffraction patterns (Supplementary Fig. 2), and polarized optical microscope images (Supplementary Fig. 3) show that azobenzene molecules **1** exhibit a crystalline phase at room temperature, a smectic LC phase between 50 and 60 °C, and an isotropic liquid phase above 60 °C.

**Ultrafast transient transmission spectroscopy**. Figure 2a shows the optical absorption spectra of *trans*-azobenzene molecules and partially photoisomerized *cis*-azobenzene molecules. The broad peak at ~300 nm is typical of the π–π* transition bands of *trans*-azobenzene molecules and decreases after *trans*–*cis* photoisomerization. In contrast, a new peak at ~450 nm appears after photoisomerization, corresponding to the n–π* transition bands of *cis*-azobenzene molecules. Thus, the peaks at approximately 300 and 450 nm decrease and increase, respectively, upon *trans*-to-*cis* photoisomerization. The ultrafast transient transmission measurements (Supplementary Fig. 4) were performed on azobenzene molecules in the LC phase (55 °C) as shown in Fig. 2b. The transient transmission spectra in the crystalline phase are shown in Supplementary Fig. 5. The wavelength of the pump light was set to 266 nm to induce *trans*-to-*cis* photoisomerization. To monitor the populations of *trans*- (π–π* transition) and *cis*-azobenzene molecules (n–π* transition), the wavelengths of the probe light were set to 266 and 400 nm. As shown in Fig. 2b, the transient transmissions at wavelengths of 266 and 400 nm increase and decrease at a delay time of 0 ps, which suggests that the populations of *trans*- and *cis*-azobenzene molecules decrease and increase, respectively, under UV photoexcitation. The fitting

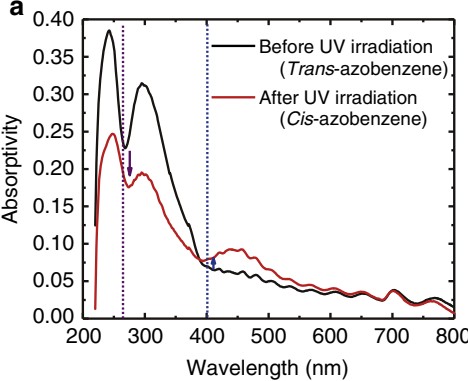

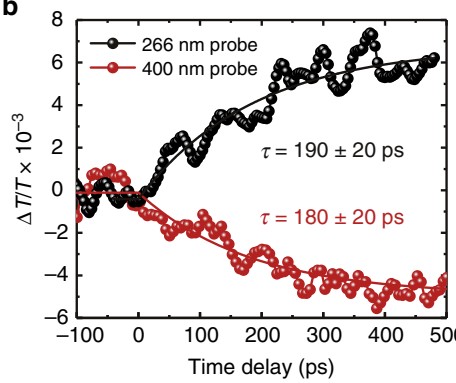

**Fig. 2** Optical absorption spectroscopy and transient transmission measurements of azobenzene molecules. **a** Optical absorption spectra of *trans*- and *cis*-azobenzene molecules. The blue and purple dashed lines indicate the absorptivity at wavelengths of 400 and 266 nm, respectively. **b** Transient transmission measurements of azobenzene molecules in the LC phase. The black and red solid lines are the fitting curve (Eq. (1)) with time constants of 190 ± 20 and 180 ± 20 ps, respectively

curves in Fig. 2b are expressed as

$$
\begin{aligned}
t < t_0 \quad & \frac{\Delta T}{T} = A_0 + y_0 \\
t \geq t_0 \quad & \frac{\Delta T}{T} = A_0 \exp\left(-\frac{t}{\tau}\right) + y_0
\end{aligned}, \quad (1)
$$

where $t$, $t_0$, $A_0$, $y_0$ and $\tau$ are the time delay, deviation at time zero, amplitude of $\Delta T/T$, offset and time constant, respectively, used as fitting parameters. Figure 2b shows that the time constants of the photoinduced responses of the azobenzene molecules in the LC phase are very slow ($\tau = 180$–190 ps). The processes of photo-isomerization can typically be separated into three parts, i.e., the electronic π–π* transition, the electronic transition to the conical intersection and the structural change from the *trans*-form to the *cis*-form. The electronic transitions occur quite rapidly (~1 ps) and the time constant of the structural change strongly depends on the surrounding conditions. The structural changes of azobenzene molecules in *n*-hexane and ethylene glycol occur in 0.5–1 ps and >10 ps, respectively[34]. The slow photoresponse observed in azobenzene molecules in the LC phase (Fig. 2b) should correspond to the structural change of the *trans*-to-*cis* photoisomerization of the azobenzene molecules in the highly viscous LC phase.

**Ultrafast time-resolved electron diffraction**. Electron diffraction measurements[32,35,36] were performed in transmission mode on an ~60-nm-thick film of azobenzene molecules. Figure 3a shows the electron diffraction pattern from the film at 52 °C, and sixfold symmetric first-order diffraction peaks are present. The $Q$-value

is defined as:

$$
Q = 4\pi\sin\theta/\lambda = 2\pi/d, \quad (2)
$$

where $\theta$, $\lambda$ and $d$ are the scattering angle, de Broglie wavelength and plane distance, respectively. The molecular orientation is visible throughout the electron beam area on the sample (a diameter of 100 μm). As shown in Fig. 3a, the $Q$-value of the diffraction spots is 1.4 ($2\pi \times 0.22$) Å$^{-1}$, which means that the plane distance is 4.5 Å. The plane distance and molecular distance have relation of $1:2\sqrt{3}$ for a hexagonal lattice; therefore, the molecular packing distance between the azobenzene molecules is calculated to be 5.1 Å, and the azobenzene molecules should be homeotropically aligned in a smectic B (hexatic) phase[37] (Fig. 3b). The intensities of the higher-order diffraction peaks are quite low. Thus, the molecules should fluctuate in their sixfold symmetric coordination. The sample thickness was determined by single-wavelength ellipsometry (Supplementary Fig. 6).

To elucidate the structural dynamics during the photoisomerization and subsequent intermolecular interactions, we performed time-resolved electron diffraction experiments, which provide direct structural information on the evolution of the periodic ordering following excitation with linearly or circularly polarized UV light ($\lambda = 266$ nm). The diffraction pattern from the azobenzene molecules in the crystalline phase did not change upon UV photoexcitation (see Supplementary Figs. 7–9 for more details), which indicates that the crystalline packing hampers the molecular motion induced by the photoisomerization of the azobenzene molecules. In contrast, UV photoexcitation modulates the sixfold symmetric diffraction pattern of the LC azobenzene molecules. Figure 4a shows differential images of the diffraction patterns before and after (−50, +20, +80, +150 and +500 ps) photoexcitation with linearly polarized UV light at 52 °C at an incident fluence ($F_I$) of 0.50 mJ cm$^{-2}$ (see Supplementary Figs. 10–14 for fluence dependence). The polarization angle of the pump laser is given by a purple arrow in the figure. The negative spots in the differential diffraction pattern represent the molecular order that was present initially and disappeared with the photoinduced molecular motion; in contrast, the positive spots represent the molecular order that was newly created upon photoexcitation. As shown in the electron diffraction patterns at a positive time delay (Fig. 4a), the diffraction spots converge at the positive signals and sixfold to rotational symmetry was observed around the diffraction spots. The snapshots of the electron diffraction show that the degree of rotation is fixed, as marked with dotted squares, even immediately after photoexcitation (+20 ps), and the intensities of the positive spots increase as time evolves. These results suggest that the sixfold symmetric molecular domains are aligned along the laser polarization axis, accompanied by the rotational motion of the molecular domains. The rotating domains connect to each other as time passes to form a higher orientation. Notably, this higher level of ordering of azobenzene molecules occurs upon photoexcitation with linearly polarized light but not with circularly polarized light (Fig. 4b and Supplementary Figs. 15 and 16). Figure 4c, d shows the evolution of the intensities of the positive and negative spots generated by photoexcitation with linearly and circularly polarized light, respectively, as functions of time. The higher molecular orientation occurs on the timescale of 170 ± 10 ps, which corresponds well to the timescale of the *trans*-to-*cis* photoisomerization observed by transient transmission spectroscopy (Fig. 2b). The intermolecular motion of the azobenzene molecules in the soft matter phase is expected to occur on longer timescales (in the range of microseconds to milliseconds); however, we found that the cooperative molecular ordering is simultaneously induced by the *trans*-to-*cis* photoisomerization of azobenzene molecules on

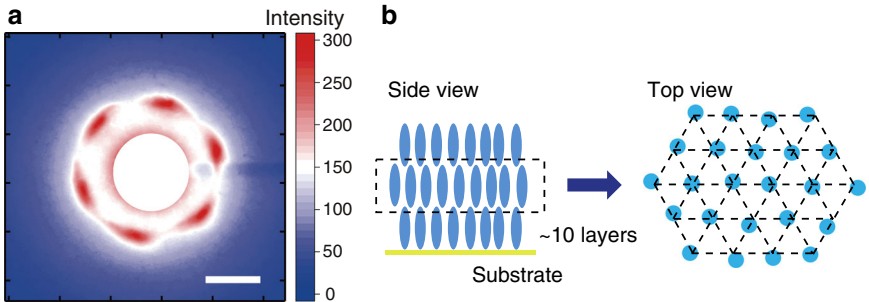

**Fig. 3** Electron diffraction image. **a** Sixfold symmetric electron diffraction pattern of LC azobenzene molecules. The scale bar for the $Q$-value (1 Å$^{-1}$) is shown as an inset. **b** The side and top views of the azobenzene molecules, which show the homeotropically aligned azobenzene molecules (side view) in a smectic B phase (top view)

the timescale of ~100 ps, which is the fastest cooperative molecular motion ever observed in an LC phase. The higher molecular orientation is also supported by the fact that the absolute change in the intensity of the positive peak is greater than that of the negative peak. The positive and negative spots in Fig. 4a are sharp and broad in shape, respectively, which also supports the idea that a higher molecular orientation occurs in LC azobenzene molecules subjected to UV photoexcitation. The reverse isomerization (cis-to-trans) occurs in a few milliseconds, since we observed the time-resolved changes at a laser repetition rate of 200 Hz (5 ms) and could not observe the changes at a repetition rate of 1 kHz (1 ms) due to signal accumulation (Supplementary Figs. 17 and 18). These major motions were uniformly induced over the entire photoexcited area (a diameter of 100 μm).

Based on the incident fluence ($F_I$ = 0.50 mJ cm$^{-2}$) of the excitation laser, the single photon energy ($E$ = 4.65 eV, $\lambda$ = 266 nm) and the absorptivity ($A$ = 23%) of the sample, the number of absorbed photons per unit area ($N_{ex}$) was calculated to be $1.5 \times 10^{14}$ photons cm$^{-2}$ using the following equation:

$$N_{ex} = (F_I A)/E \qquad (3)$$

The number of molecules per unit area ($N_{mol}$) was determined by the sample thickness ($d$ = ~60 nm), volume density ($\rho$), molecular weight ($m$ = 518.86 g mol$^{-1}$) and Avogadro constant ($N_A$) according to the following equation:

$$N_{mol} = (d \rho N_A)/m \qquad (4)$$

The typical volume density value of LC materials is ~1 g cm$^{-3}$; therefore, the number of molecules per unit area is calculated to be ~$7 \times 10^{15}$ molecules cm$^{-2}$. Thus, the ratio of bent molecules ($\rho_{cis}$) is calculated to be ~1% by the following equation:

$$\rho_{cis} = N_{ex}/N_{mol} \times \eta = 1\%, \qquad (5)$$

where $\eta_i$ is the photoisomerization quantum efficiency (50%)[38–40]. The ability of that such a small fraction of cis-azobenzene molecules within the predominantly trans-azobenzene molecules to induce a higher orientation was surprising. The trans-to-cis isomerization was expected to disorder the system; however, the opposite was observed, i.e., a small fraction of cis-azobenzene molecules oriented in the same direction rotate the molecular system and induce a higher orientation through the very nature of the molecular interactions in the LC phase. The change in the intensities of the positive spots observed in Fig. 4c is 12%. Thus, the sample became ~10% more oriented upon the photoisomerization of 1% of the azobenzene molecules.

**Molecular dynamics calculations**. This proposed mechanism of the molecular orientation of the LC azobenzene molecules can be explained by molecular dynamics calculations[41,42]. Figure 5a, b shows the model LC azobenzene molecules assembled with and without a bent azobenzene molecule, respectively, and their calculated electron diffraction patterns. The azobenzene molecules are loosely packed with quasi-sixfold symmetry. A fraction of the azobenzene molecules were photoisomerized with linearly polarized light (Fig. 5a). Notably, the photoisomerized molecules have a transition dipole moment parallel to the polarization of the incident light. Approximately one percent of the azobenzene molecules undergo trans-to-cis photoisomerization via UV photoexcitation; for example, the molecules shown in an ellipse in Fig. 5a undergo photoisomerization and are converted to the cis-form with a bent structure. The bent molecules push against the adjacent molecules due to steric forces, which decreases the free space in the plane in the direction of the bend (laser polarization axis). This motion rotates and aligns the primary and/or secondary neighbouring molecules into the sixfold symmetric pattern. Because of the lower free space in the plane, the molecular system with bent molecules (Fig. 5a) has a greater degree of orientation than the initial loosely packed system (Fig. 5b), as represented by the dotted black hexagons. The direction of the rotation in the system depends on the initial sixfold symmetric coordination and the laser polarization axis (Supplementary Note 2, Supplementary Figs. 19–21). Further details on the density dependence of the diffraction pattern calculated by molecular dynamics and statistical analyses are given in Supplementary Notes 3 and 4, and Supplementary Figs. 22–31. The trans-to-cis isomerization of the azobenzene molecules occurs uniformly towards the same direction over the entire photoexcited area, accelerating the intermolecular motions. In contrast, circularly polarized light bends the molecules in various directions, which does not result in uniform intermolecular motion.

## Discussion

Ultrafast time-resolved electron diffraction was used to monitor the structural dynamics of representative LC azobenzene molecules. Photoexcitation with UV light triggers the trans-to-cis isomerization of LC azobenzene molecules in ~100 ps. The simultaneous and coherent change in the azobenzene molecules and subsequent steric interactions induce ultrafast higher molecular orientation in the LC azobenzene molecules. The observed ultrafast phenomena indicate that the cooperative motion of the molecules in the LC phase is induced by the intermolecular interactions in the sixfold symmetric

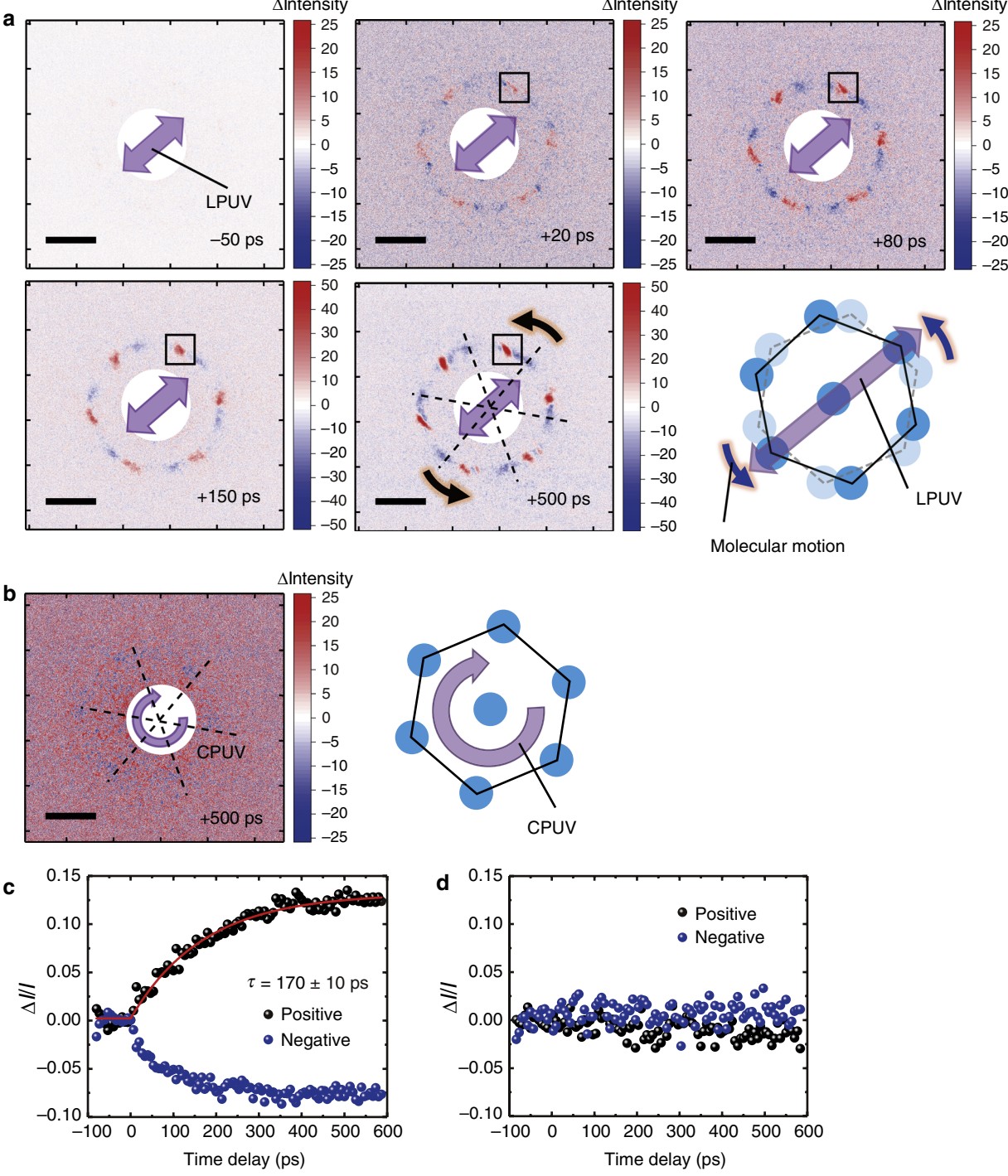

**Fig. 4** Ultrafast time-resolved electron diffraction measurements. **a** Snapshots of differential diffraction patterns before and after excitation with linearly polarized UV light at −50, +20, +80, +150 and +500 ps. Schematic illustrations of the phenomena obtained from the differential diffraction patterns are inset in the figures. The black dotted squares represent the same position showing representative positive spots. The black dashed lines indicate the initial diffraction spots to guide the eyes. The black arrows indicate the motion direction of the diffraction spots. **b** Differential diffraction patterns at a delay time of +500 ps after excitation with circularly polarized UV (CPUV) light with inset of schematic illustrations describing the phenomena. The scale bar for the Q-value (1 Å$^{-1}$) is also shown as an inset. **c, d** The intensity changes in positive and negative electron diffraction spots generated by photoexcitation with linearly and circularly polarized light, respectively, as functions of time. The red solid line (170 ± 10 ps) uses the same function as the fitting curves shown in Fig. 2b

coordinated matrix. The series of ultrafast intra- and inter-molecular motions initiated by photoexcitation introduces a new direction in the fundamental chemistry of photoresponsive molecules in soft matter as well as ultrafast soft-actuator applications of azobenzene molecules. The methodology used in this ultrafast observation is also important because it includes a general conceptual advance for elucidating the structural changes in stimulus-responsive units in even more complex soft materials[43–46], e.g., functional LC polymers and synthetic cell membranes.

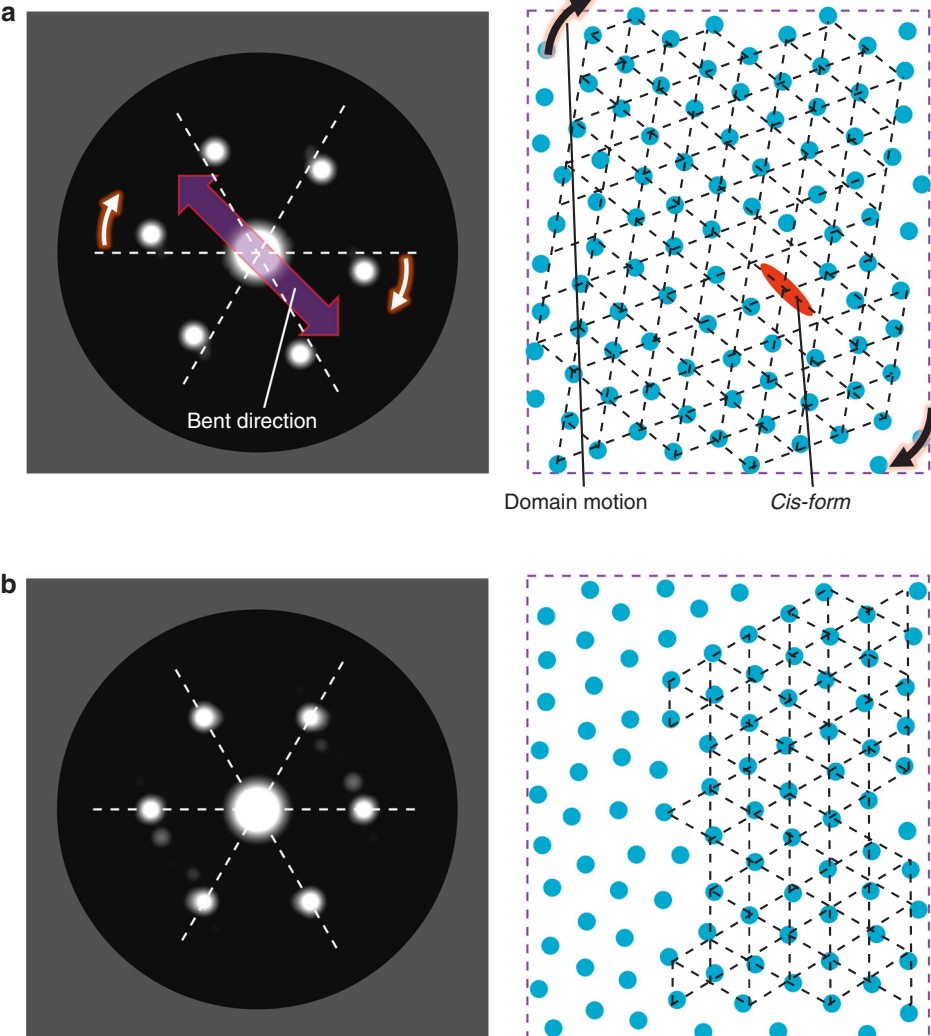

**Fig. 5** Molecular dynamics simulations. **a** Calculated diffraction pattern (left) from the coordinates of a *trans*-azobenzene molecule and a *cis*-azobenzene molecule (right). The white and black arrows indicate the motion directions of the diffraction spots and molecules, respectively. The purple arrow indicates the bending direction of the *cis*- molecule. **b** Calculated diffraction pattern (left) from the coordinates of the *trans*-azobenzene molecules. **a**, **b** correspond to the azobenzene molecules in an LC assembly after and before UV photoexcitation, respectively. The white dashed lines guide the eyes

## Methods

**Optical spectroscopy of azobenzene molecules**. Azobenzene molecules in toluene solution (2 w/w%) were spin-coated on bulk sapphire substrates at 2000 rpm. The resulting thin-film sample was heated and melted on a hotplate at a temperature of 80 °C in 3 min; the temperature of the surface of the hotplate was measured with a K-type thermocouple. The sample was quenched to room temperature, and the static optical spectrum (transmission and reflection) of the sample (*trans*-azobenzene molecules) was measured by a commercial spectrometer (JASCO V-670) at room temperature. The same sample was subsequently heated and melted on a hotplate at 80 °C while being continuously irradiated with UV light (~20 W m$^{-2}$) for 3 min. The sample was quenched to room temperature while still being irradiated with irradiated UV light, and the static optical spectrum of the sample (*cis*-azobenzene molecules) was measured by the spectrometer at room temperature. This procedure was performed because the sample does not undergo *trans–cis* or *cis–trans* isomerization in the crystalline phase, but only in the liquid or LC phase. The optical spectra of *trans*- and *cis*-azobenzene molecules are shown in Fig. 2a.

**Transient transmission spectroscopy**. Transient transmission spectroscopy was performed on the azobenzene molecules in the LC (55 °C) phases in air. The experimental setup is shown in the supplementary information (Supplementary Fig. 4). The wavelength of the pump pulse was set to 266 nm and that of the probe pulse to 266 or 400 nm. The pump and probe optical pulses were focused on the sample, and the transmitted visible probe light was detected with a GaP photodiode. The incident fluence of the pump light was 0.50 mJ cm$^{-2}$ at a repetition rate of 250 Hz. The diameter of the pump light was measured to be 400 μm with a knife edge. The repetition rate of the probe light was 500 Hz, and the modulation of the

probe light to 250 Hz (the repetition rate of the pump light) was obtained by a lock-in amplifier. A solution of the sample in toluene (2 w/w%) was spin-coated onto a bulk sapphire substrate.

**Time-resolved electron diffraction**. The experimental setup of the compact DC-accelerated electron diffraction system is provided elsewhere[32,35]. UV pulses were focused on the sample film. The incident laser fluence was 0.15–0.50 mJ cm$^{-2}$, and the acceleration voltage of the probe electron pulses was 75 keV under a DC electric field. The photoinduced structural changes within the material were investigated with an electron pulse containing $2 \times 10^4$ electrons confined to a 100-μm-diameter spot. The pulse duration of the electron beam was >1 ps. To avoid the accumulation of photoisomerization, the repetition rate was set to 200–333 Hz depending on the incident laser fluence. The sample was spin-coated on SiN (30-nm-thick) membranes.

**Molecular dynamics calculation**. Two-dimensional molecular dynamics calculations were performed based on the LAMMPS Molecular Dynamics Simulator[47]. As indicated in ref. [16], the correlation among the interlayer molecules is much weaker than that among the intralayer molecules in the smectic B (Hex) phase; therefore, two-dimensional MD simulation should be appropriate. Azobenzene molecules were simplified to spheroids in the model, and the ratios of the three axes of the simplified *trans*- and *cis*-azobenzene molecules were 1:1:10 and 1:3:8, respectively (Supplementary Fig. 22). The potential energies of the azobenzene molecules were generated from the standard Gay-Berne model[41,42]. The normalized density of the molecules was used as a parameter (0.96, 0.97 and 0.98), and the density of the close-packed condition was set to 1.00. One hundred azobenzene molecules were

contained in a unit system, and photoexcitation could change an azobenzene molecule from the straight form to the bent form (1%). The calculated diffraction pattern was generated using CrystalMaker® and SingleCrystal® software[48] using molecular coordinates obtained from the molecular dynamics calculations.

## Data availability

The source data underlying Fig. 2a, b, 3a, 4a (−50, +20, +80, +150, and +500 ps)−d, 5a and b and Supplementary Figs 1, 2, 5, 6, 7a−c, 8a, b, 9, 10a, b, 11, 12a, b, 13, 14, 15a, b, 16, 17a, b, 18–21, 23, 24a, b, 25, 26a, b, 27, 28a, b, 30, and 31 are provided as a Source Data file. Other relevant data are available from the corresponding authors upon reasonable request.

## Code availability

All data analyses were performed using publicly available software.

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

## Acknowledgements

This work was supported by JST, PRESTO (Grant number JPMJPR13KD) and JSPS KAKENHI (Grant numbers JP18H04519 and JP18H05208). This work was also partially supported by JST, CREST (Grant number JPMJCR1422). M.H. is grateful for the support from the JSPS Leading Initiative for Excellent Young Researchers. D.Y. is grateful for financial support from the JSPS Research Fellowship for Young Scientists. We would like to thank Prof. Takayoshi Yokoya at Okayama University and Prof. Jiro Matsuo and Prof. Toshio Seki at Kyoto University for the experimental support.

## Author contributions

M.H. and T.K. conceived the idea and designed the study. D.Y. carried out the synthesis and static measurements. M.H., T.I. and S.K. performed the transient transmission experiments. M.H., T.S. and Y.H. performed the ultrafast time-resolved measurements. K.T. and M.H. performed the molecular dynamics calculations. M.H., D.Y., K.I. and T.K. interpreted the data. M.H., D.Y. and T.K. wrote the paper. All authors discussed the results and edited the paper.

## Additional information

**Competing interests:** The authors declare no competing interests.

