## [Peer Review File · Nature Communications]

Reviewers' comments:

Reviewer #1 (Remarks to the Author):

The study of Hada et al is of great experimental quality and I believe it deserves publication in a high impact peer-reviewed journal such as Nature Communications. The effect of producing a more ordered LC through the generation of "defects" (bent azobenzene molecules due to photo induced trans-to-cis isomerization) is not obvious and will be of significant interest to a wide variety of scientists from ultrafast structural dynamics, soft matter, and materials science and engineering communities.

There are, however, several points which have to be clarified/corrected before this article is accepted for publication.

1. In the introduction section the authors claim that this study is the very first of its kind to observe isomerization dynamics by direct means, i.e. through ultrafast electron diffraction: "to the best of our knowledge, neither the ultrafast structural dynamics of the photoisomerization of molecules nor the subsequent responses have been directly observed." This is definitely not the case. Consider, for instance, UED experiments on photoinduced ring-closing reaction in diarylethene performed by Jean-Ruel and coworkers, *J. Phys. Chem. B* 2013, 117, 15894–15902.

2. "The molecules bend towards the light polarization direction, which leads to an overall molecular alignment in this direction."

Please be "more scientific" when explaining bending direction of the molecules, i.e. consider transition dipole moment of the azobenzene chromophore and its relation to the molecular structure; mention that azobenzene "disks" are randomly oriented in LC; the only molecules that get photoexcited are those with transition dipole moment along the laser polarization axis; this forms a pre-selected ensemble of molecules bending towards a particular direction, which in turn results in anisotropic squeezing of other molecules in LC and ultimately in statistically higher ordering of the sample. The text, however, clearly states "an overall molecular alignment in this direction" meaning that the molecules are getting better aligned only in the direction of laser polarization. This is not clearly evident from the MD simulation snapshots.

3. It is hard to judge about better degree of molecular ordering based only on visual inspection of the snapshot from MD simulations. The authors could stronger support their argument about higher molecular ordering/disordering in the sample through the statistical analysis of molecular coordinates. That is to establish some sort of an index related to the deviation of molecules from the ideal hexagonal packing. Once automated such analysis can be ran on multiple snapshots for excited/unexcited sample to get better statistical representation.

4. Cumulative signal (main text & supplemental): "... could not observe the changes at a repetition rate of 1 kHz (1 ms) due to signal accumulation (Figs. S17 and S18)." My expectations would be a slowly decaying high-level background signal for negative time-delays and some signal evolution (most likely also with a significant background and noise) for positive time-delays. This prediction is, however, based on a conventional data acquisition scheme employed in most UED experiments, i.e. the diffraction images with pump on/off are collected over multiple electron laser shots, so the molecules have time to fully relax before the pump shutter opens (or the blank chopper sector moves away). Could the authors comment why they did not get any net intensity changes at positive time-delays even when cumulative effect was present?

5. Equation 1 cannot contain elementary charge. Yes, the energy of photon has to be converted from eV to J, so one got to use the conversion factor, which numerically (due to definition) is equal to that of an electron BUT (!!!) has units of J/eV. To avoid any further confusion, I suggest to remove "e" from Eq. 1.

6. "The change in the intensities of the positive spots observed in Fig. 3e is 12%; therefore, a photoisomerized LC azobenzene molecule should influence the alignment the 12 adjacent molecules via UV-photoexcitation." Could the authors justify how/why 12% of intensity increase directly translates to the same number of better aligned molecules? Maybe it is better to state that the sample became about 10% more ordered? (page 9)

7. The authors can provide an additional argument towards higher degree of alignment upon

photoexcitation with linearly UV light by highlighting the fact that "positive spots" are also better defined in shape (Fig 3a).

8. Change text in captions for Figures S19-S21 to explain why spots rotate clockwise/counterclockwise or none.

9. There are a lot of nicely looking simulated diffraction patterns in Figures S25, S27 without much of explanations what the reader should pay attention for.

Minor corrections. Several statements have to be either rephrased for clarity or corrected

1. "ultrafast higher orientation to a single direction", "Azobenene " (page 3)

2. "We found as trans-to-cis isomerization in the azobenzene-based LC assembly via a linearly polarized ultraviolet (UV) light induced ultrafast higher orientation." (page 4)

3. "The molecular orientation is visible throughout the observing area ($\phi 100 \mu\text{m}$). " A bit of an awkward statement. I think the authors meant the photoexcitation or electron beam area on the sample. (page 6)

4. "As shown in Fig. 3a, the diffraction spots converge at the positive signals and six-fold to rotational symmetry was observed around the diffraction spots." My guess is that the authors originally meant "... diffraction spots emerge as a positive net intensity signal with six-fold rotational symmetry" but somehow the sentence got scrambled. Likewise, "... aligned along the direction of the polarized angle of the incident laser accompanied ... " perhaps means "... aligned {along/in the direction} of the laser polarization axis ... " Please have another read through the paper to clean up the text. (page 7)

5. "Thus, a photoisomerization quantum efficiency (η) of $\sim 50\%$ results in approximately 1% of the molecules being excited. " is misleading. From the context of the paper it seems like 1% should really refer to the bent azobenzene molecules, i.e. Total number of irradiated species \times Excitation QE \times Photoisomerization QE = 1% of bent molecules. (page 8)

6. "A few percent of the azobenzene molecules undergo trans-to-cis photoisomerization via UV-photoexcitation;" does not match to previously estimated 1% of isomerized molecules. (page 9)

7. "Because of the lower free volume in the plane ... " Change "volume" to "space" as there is no volume in plain. (page 10)

Reviewer #2 (Remarks to the Author):

In the manuscript "Ultrafast Isomerization-Induced Cooperative Motions to Higher Molecular Orientation in Smectic Liquid-Crystalline Azobenzene Molecules", the author observed the formation of a higher molecular orientation state upon UV excitation of an azobenzene derivative in liquid crystal phase, on the timescale of ~ 100 ps. Their conclusion is supported by ultrafast electron diffraction (UED) data and transient absorption/transmission data. The main claim is based on the UED data displayed in Fig. 3a and is supported by a simplified MD calculation. Overall speaking, the data, simulation and data interpretation is sufficient to support the claim that a higher molecular orientation state is formed in ~ 100 ps, and such a large-scale motion is induced by the trans-cis isomerization of a small subset of individual molecules. The conclusion and methodology could be interesting to the relevant communities. However, in order to make their claims convincing, several critical weak points still need to be addressed.

1. The claim "cooperative motion" is not supported by any experimental data or simulation.

In the current manuscript, an important claim is "cooperative motion", which appeared in the title, abstract, and multiple times in the text. This claim, however, is not supported by any data. The UED data essentially show an "initial state" (6-fold symmetry) and a "final state" (rotated with respect to initial state and a higher orientation order), but did not show what's the actual motion looks like (how it gets from initial to final state). For example, it could be a concerted large-scale rotation, or cascaded motion (nearest neighbors to 2nd nearest neighbors to 3rd nearest neighbors and so on), or layer-by-layer motion, etc. The author seems to indicate it's a concerted large-scale rotation by arrows in Fig. 3 and Fig. 4, but not really supported by any data. In addition, the MD simulation is only performed to two equilibrium states, not to any non-equilibrium state. Therefore

the current simulation also cannot reveal any detailed dynamics. The authors need to either provide evidence to "cooperative motion" or revoke this claim.

2. More information is needed on the transient absorption data. I believe the transient absorption data is critical to the storyline because the appearance of the $n-n^*$ band is the only evidence of the occurrence of the isomerization. The transient transmission data show a decay but is less convincing to conclude that this is isomerization. The author needs to provide more information on the transient absorption measurement. How is it measured? What's the pump-probe delay, fluence, pulse duration etc. Now the figure legend says "UV light 3 min" without any explanation what it means. This is really confusing because a pump-probe delay of 3 minutes is contradicting that the original phase is recovered within 5 ms after pump. In addition, considering the importance of spectroscopic studies to the storyline I suggest moving Fig S4 and S5 to the main text.

3. The MD simulation needs to be better explained. The authors said the MD simulation is "2D", does it mean only a single layer is considered? The data show a 12% change which clearly means more than one layer is affected, and therefore a single-layer model might not be sufficient. In addition, Fig. S22 to S27 has very little description text, and it's not obvious what information the authors are trying to convey.

4. The authors concluded "the trans-to-cis photoisomerization occurs on the timescale of ~ 110 ps" from the transient transmission data, but it is questionable. The transient transmission data show a ~ 110 ps dynamics, but why is this not the same dynamics as observed in UED, which occurs at a very similar timescale? I believe the photoisomerization itself should be very fast (typically sub-picosecond) because it's an intramolecular motion, and the subsequent large-scale re-orientation could be much slower (~ 100 ps), because it is induced by intermolecular interaction. The authors should be more careful when describing photoisomerization vs higher orientation.

Above are my main concerns, there are some minor issues also need to be addressed:

1. The author uses "azobenzene molecule" or "azobenzene" to represent azobenzene derivative throughout the text. It is not a big problem but it's still good to clarify at some point in the text.

2. The definition of Q. In the UED community it is a convention that Q represents momentum transfer ($Q=4\pi/\lambda*\sin(\theta/2)$), and I believe there's a factor of 4π from the author's definition. It needs to be clarified somewhere.

3. Fig. 2a is missing color bar.

4. The text "the molecular packing distance between the azobenzene molecules was 5.1 \AA , and the azobenzene molecules should be homeotropically aligned and resides in a smectic B (hexatic) phase" needs some reference. Where was the number 5.1 \AA coming from? How did the authors conclude that the azobenzene molecules are in this phase?

5. Fig. 3a and 3b should have something to show the original Bragg peak direction, something like the white dashed lines in Fig. 4 will be good.

6. Fig. S9 shows the UED for crystalline phase sample, which essentially nothing changed after time zero. However, there should at least be some heating effect (Debye-Waller), since the crystalline sample has comparable absorbance at 266 nm as the LC sample. Can the authors explain why there's no Debye Waller effect?

7. There are many typos in the manuscript:

-Page 5, Fig. S5 and S6 should be S4 and S5

-Page 9, there's no Fig. 3e

-Page 11, "Conventional transient absorption spectroscopy" should be "Conventional transient transmission spectroscopy".

In sum, I cannot support the manuscript to be published in Nature Communications, at least in its current form.

We thank the reviewers for giving us fruitful comments about our manuscript. We carefully revised our manuscript according to the comments. The responses to each reviewer are as follows.

Reply to Reviewer: 1

The study of Hada et al is of great experimental quality and I believe it deserves publication in a high impact peer-reviewed journal such as Nature Communications. The effect of producing a more ordered LC through the generation of “defects” (bent azobenzene molecules due to photo induced trans-to-cis isomerization) is not obvious and will be of significant interest to a wide variety of scientists from ultrafast structural dynamics, soft matter, and materials science and engineering communities. There are, however, several points which have to be clarified/corrected before this article is accepted for publication.

We appreciate his/her recommendation of our work for acceptance. We also thank him/her for the many fruitful suggestions to improve the quality of the manuscript. The answers to the comments are appended below.

1. In the introduction section the authors claim that this study is the very first of its kind to observe isomerization dynamics by direct means, i.e. through ultrafast electron diffraction: “to the best of our knowledge, neither the ultrafast structural dynamics of the photoisomerization of molecules nor the subsequent responses have been directly observed.” This is definitely not the case. Consider, for instance, UED experiments on photoinduced ring-closing reaction in diarylethene performed by Jean-Ruel and coworkers, J. Phys. Chem. B 2013, 117, 15894–15902.

As the reviewer indicated, the ultrafast ring-closing reaction of diarylethene in the crystal phase was observed using time-resolved electron diffraction by H. Jean-Ruel and coworkers (J. Phys. Chem. B 2013, 117, 15894–15902), which was cited as Ref. 31. Our study is the first structural dynamics observation of the ultrafast photoisomerization of azobenzene molecules in the liquid-crystalline phase.

We have also revised the manuscript as “*However, to the best of our knowledge, neither the ultrafast structural dynamics of the photoisomerization of molecules in the LC phase nor the subsequent responses have been directly observed.*” (page 3 yellow marked)

2. “The molecules bend towards the light polarization direction, which leads to an overall molecular alignment in this direction.” Please be “more scientific” when explaining bending direction of the molecules, i.e. consider transition dipole moment of the azobenzene chromophore and its relation to the molecular structure; mention that azobenzene “disks” are randomly oriented in LC; the only molecules that get photoexcited are those with transition dipole moment along the laser polarization axis; this forms a pre-selected ensemble of molecules bending towards a particular direction, which in turn results in anisotropic squeezing of other molecules in LC and

ultimately in statistically higher ordering of the sample. The text, however, clearly states “an overall molecular alignment in this direction” meaning that the molecules are getting better aligned only in the direction of laser polarization. This is not clearly evident from the MD simulation snapshots.

We agree with the reviewer’s comments. The photoisomerization of azobenzene molecules occurs to the specific direction that corresponds to the direction of the transition dipole moment of the azobenzene molecules and laser polarization axis because of the selective excitation of the azobenzene molecules with the direction of the transition dipole moment corresponding to the laser polarization axis, where the transition dipole moments of azobenzene “disks” are randomly oriented in the LC phase. This selective isomerization and subsequent steric effect to one direction induces the molecular alignment (higher orientation). The molecular alignment to the laser polarization axis can be understood by the ultrafast time-resolved electron diffraction, and the molecular alignment to the bending direction of azobenzene molecules can be understood by the molecular dynamics simulations. We conclude that the azobenzene molecules align to the molecular bending direction (corresponding to the transition dipole moment and laser polarization axis) by UV photoexcitation, which is revealed by the combination of the experiments and calculations. The flexibility or fluidity of the liquid crystal works effectively on this phenomenon. We have revised the manuscript accordingly. (page 5 yellow marked)

3. It is hard to judge about better degree of molecular ordering based only on visual inspection of the snapshot from MD simulations. The authors could stronger support their argument about higher molecular ordering/disordering in the sample through the statistical analysis of molecular coordinates. That is to establish some sort of an index related to the deviation of molecules from the ideal hexagonal packing. Once automated such analysis can be ran on multiple snapshots for excited/unexcited sample to get better statistical representation.

This comment is related to major comment #3 by reviewer #2. We have performed a statistical analysis (bond-order analysis) of the molecular dynamics (MD) simulation. Following Chaikin, P. M. & Lubensky, T. C. in Principles of Condensed Matter Physics (Cambridge Univ. Press 1995), we defined the bond-order parameter as $\varphi(\eta) = |\langle \exp(i\eta\theta) \rangle|^2$, where η and θ are the symmetry of the system and angle among three arbitrary molecules respectively. As shown in Fig. S30, the ideal hexagonal lattice represents a single peak in $\varphi(\eta)$ at $\eta = 6$ (six-fold symmetry). Figure S31 shows the bond-order parameters with and without a bent molecule from MD simulations (density: 0.98). As shown in the figures, the bond-order parameter with a bent molecule is approximately 50% higher than that without a bent molecule. This tendency agrees with the time-resolved electron diffraction measurements, *i.e.*, the intensity of the electron diffraction spots increases by $\sim 12\%$ by photoexcitation. We have added this analysis as supplementary information.

4. Cumulative signal (main text & supplemental): “... could not observe the changes at a repetition rate of 1 kHz (1 ms) due to signal accumulation (Figs. S17 and S18).” My expectations would be a slowly decaying high-level background signal for negative time-delays and some signal evolution (most likely also with a significant background and noise) for positive time-delays. This prediction is, however, based on a

conventional data acquisition scheme employed in most UED experiments, i.e. the diffraction images with pump on/off are collected over multiple electron laser shots, so the molecules have time to fully relax before the pump shutter opens (or the blank chopper sector moves away). Could the authors comment why they did not get any net intensity changes at positive time-delays even when cumulative effect was present?

As the reviewer expected, this experiment is also based on the conventional data acquisition scheme employed in most UED experiments (diffraction images with pump on/off collected over multiple electron laser shots). The cumulative effect of the UED is shown in Gao, M. et al. Mapping molecular motions leading to charge delocalization with ultrabright electrons. *Nature* **496**, 343-346 (2013). We have added the relevant reference to the supplementary information. When the cumulative effect is present in our case, the system has a significant number of *cis*-azobenzene molecules before photoexcitation. The higher orientation of the azobenzene molecules in the liquid-crystalline phase occurs because a small number of bent molecules emerge in the majority of straight molecules. If the *cis*-azobenzene accumulates, this effect would not be observed.

5. Equation 1 cannot contain elementary charge. Yes, the energy of photon has to be converted from eV to J, so one got to use the conversion factor, which numerically (due to definition) is equal to that of an electron BUT (!!!) has units of J/eV. To avoid any further confusion, I suggest to remove “e” from Eq. 1.

We appreciate the comment from the reviewer. We have removed the “e” from revised Eq. 2. (page 11)

6. “The change in the intensities of the positive spots observed in Fig. 3e is 12%; therefore, a photoisomerized LC azobenzene molecule should influence the alignment the 12 adjacent molecules via UV-photoexcitation.” Could the authors justify how/why 12% of intensity increase directly translates to the same number of better aligned molecules? Maybe it is better to state that the sample became about 10% more ordered? (page 9)

We agree with the reviewer’s comment. This comment is also related to other comments (reviewer #1 minor #6 and reviewer #2 major #1). The ratio of the azobenzene molecules undergoing the photoisomerization is calculated to be only 1%. The motions of 1% of the azobenzene molecules increase the intensity of the electron diffraction spot by 12%. The change in the intensity of the electron diffraction spot by 12% does not simply correspond to the change of 12% of the molecules. Thus, we should state that the sample became about 10% more oriented by the photoisomerization of 1% of the azobenzene molecules.

We have revised the manuscript as “*The change in the intensities of the positive spots observed in Fig. 4c is 12%. Thus, the sample became approximately 10% more oriented by the photoisomerization of 1% of the azobenzene molecules.*”. (page 11 yellow marked)

7. The authors can provide an additional argument towards higher degree of alignment upon photoexcitation with linearly UV light by highlighting the fact that “positive spots” are also better defined in shape (Fig 3a).

We thank the reviewer for the suggestion. The shapes of the positive and negative spots are important to make our claim more convincing. The positive spots are sharp in shape, while the negative spots are broader in shape, which also suggests the higher orientation after photoexcitation. We have compared the shapes of the positive and negative spots in the main text as “*The positive and negative spots in Fig. 4a are sharp and broad in shape, respectively, which also supports the idea that a higher molecular orientation occurs in LC azobenzene molecules subjected to UV photoexcitation.*”. (page 10 yellow marked)

8. Change text in captions for Figures S19-S21 to explain why spots rotate clockwise/counterclockwise or none.

The direction in which the molecules are aligned is determined by the direction of the laser polarization axis; however, the rotational direction which the molecules are aligned is determined by both the direction of the laser polarization axis and the initial six-fold symmetric coordination of the azobenzene molecules. The laser polarization axis is fixed in the ultrafast time-resolved electron diffraction experiments, but the initial six-fold symmetric coordination is random because the sample is fabricated via spin-coating. Therefore, we can observe the rotation both clockwise and counterclockwise. If the six-fold symmetric coordination is initially aligned with the laser polarization axis, the rotation does not occur. We have added this explanation in the supplementary information.

9. There are a lot of nicely looking simulated diffraction patterns in Figures S25, S27 without much of explanations what the reader should pay attention for.

This comment is related to major comment #3 from reviewer #2. We have added explanations for Figs. S23–S28 in the supplementary information. The two-dimensional MD simulation was performed on azobenzene molecules at various densities (0.96, 0.97, and 0.98). The condition of the density at 1.00 corresponds to close packing. The azobenzene molecule is ellipsoidal, and the sizes of the *trans*- and *cis*-azobenzene molecules are 1:1:10 (straight form) and 1:3:8 (bent form), respectively. One hundred azobenzene molecules are contained in a unit system, and the photoexcitation can change an azobenzene molecule from the straight form to the bent form (1%). The azobenzene molecules in the liquid-crystalline phase before and after the photoexcitation are represented by the systems without and with a bent molecule, respectively. Figures S24, S26, and S28 show 20 calculated electron diffraction patterns (on the timescale of 5–100 LJt) at densities of 0.96, 0.97, and 0.98, respectively. As shown in Fig. S28, the six-fold symmetry of the electron diffraction patterns increases in the presence of a bent molecule at a higher density (0.98), *i.e.*, the azobenzene molecules in the liquid-crystalline phase become more oriented with the addition of bent molecules.

Minor corrections. Several statements have to be either rephrased for clarity or corrected

We appreciate the reviewer’s comments and suggestions and we have revised our manuscript accordingly.

1. “ultrafast higher orientation to a single direction”, “Azobenene ” (page 3)

“Azobenene” → “Azobenzene” (page 3 yellow marked)

2. *“We found as trans-to-cis isomerization in the azobenzene-based LC assembly via a linearly polarized ultraviolet (UV) light induced ultrafast higher orientation.” (page 4)*

“as” → “that” (page 4 yellow marked)

3. *“The molecular orientation is visible throughout the observing area ($\phi 100 \mu\text{m}$).” A bit of an awkward statement. I think the authors meant the photoexcitation or electron beam area on the sample. (page 6)*

“observing area” → “electron beam area on the sample” (page 8 yellow marked)

4. *“As shown in Fig. 3a, the diffraction spots converge at the positive signals and six-fold to rational symmetry was observed around the diffraction spots.” My guess is that the authors originally meant “ ... diffraction spots emerge as a positive net intensity signal with six-fold rotational symmetry” but somehow the sentence got scrambled. Likewise, “... aligned along the direction of the polarized angle of the incident laser accompanied ... ” perhaps means “ .. aligned {along/in the direction} of the laser polarization axis ... “ Please have another read through the paper to clean up the text. (page 7)*

“aligned along the direction of the polarized angle of the incident laser” → “aligned along the laser polarization axis” (pages 2 and 12 and caption of Fig. 1 yellow marked)

5. *“Thus, a photoisomerization quantum efficiency (η) of ~50% results in approximately 1% of the molecules being excited. ” is misleading. From the context of the paper it seems like 1% should really refer to the bent azobenzene molecules, i.e. Total number of irradiated species x Excitation QE x Photoisomerization QE = 1% of bent molecules. (page 8)*

We thank the reviewer and have revised it accordingly. (page 11 yellow marked)

6. *“A few percent of the azobenzene molecules undergo trans-to-cis photoisomerization via UV-photoexcitation;” does not match to previously estimated 1% of isomerized molecules. (page 9)*

“A few percent” → “Approximately one percent” (page 12 yellow marked)

7. *“Because of the lower free volume in the plane ... ” Change “volume” to “space” as there is no volume in plain. (page 10)*

We revised “volume” to “space” since it is based on a plane. (page 12 yellow marked)

Reply to Reviewer: 2

In the manuscript “Ultrafast Isomerization-Induced Cooperative Motions to Higher Molecular Orientation in Smectic Liquid-Crystalline Azobenzene Molecules”, the author observed the formation of a higher molecular orientation state upon UV excitation of an azobenzene derivative in liquid crystal phase, on the timescale of ~100 ps. Their conclusion is supported by ultrafast electron diffraction (UED) data and transient absorption/transmission data. The main claim is based on the UED data displayed in Fig. 3a and is supported by a simplified MD calculation.

Overall speaking, the data, simulation and data interpretation is sufficient to support the claim that a higher molecular orientation state is formed in ~100 ps, and such a large-scale motion is induced by the trans-cis isomerization of a small subset of individual molecules. The conclusion and methodology could be interesting to the relevant communities. However, in order to make their claims convincing, several critical weak points still need to be addressed.

We are grateful that the reviewer highly values our work. We also thank him/her for the many fruitful suggestions to improve the quality of the manuscript. The answers to his/her comments are appended below.

1. The claim “cooperative motion” is not supported by any experimental data or simulation. In the current manuscript, an important claim is “cooperative motion”, which appeared in the title, abstract, and multiple times in the text. This claim, however, is not supported by any data. The UED data essentially show an “initial state” (6-fold symmetry) and a “final state” (rotated with respect to initial state and a higher orientation order), but did not show what’s the actual motion looks like (how it gets from initial to final state). For example, it could be a concerted large-scale rotation, or cascaded motion (nearest neighbors to 2nd nearest neighbors to 3rd nearest neighbors and so on), or layer-by-layer motion, etc. The author seems to indicate it’s a concerted large-scale rotation by arrows in Fig. 3 and Fig. 4, but not really supported by any data. In addition, the MD simulation is only performed to two equilibrium states, not to any non-equilibrium state. Therefore the current simulation also cannot reveal any detailed dynamics. The authors need to either provide evidence to “cooperative motion” or revoke this claim.

This comment is related to major comment #6 of reviewer #1. Regarding the first part of the comment, the “cooperative motion” is represented by the fact that the motions of 1% of the azobenzene molecules change the intensity of the electron diffraction by 12%. The ratio of the azobenzene molecules undergoing the photoisomerization is calculated to be only 1%. The motions of 1% of the azobenzene molecules increase the intensity of the electron diffraction spot by 12%. The change in the intensity of the electron diffraction spot by 12% does not simply correspond to the change of 12% of the molecules. However, the point defects linearly influence the intensity of the electron diffraction, *i.e.*, the motions of 1% of the azobenzene molecules are thought to increase the intensity of the diffraction pattern by approximately 1% without any “cooperative motion”. The photoisomerized molecules push the neighbouring molecules and second neighbouring molecules by steric effects. This kind of cooperative

molecular motion in the liquid-crystalline phase has been previously reported (Barrett, C. J. et al. *Soft Matter*, 3, 1249–1261 (2007) and Ichimura, K. et al. *Langmuir* 4, 1214–1216 (1988), which were cited as Refs 2 and 239. (page 11 yellow marked)

Regarding the latter part of this comment, we totally agree with the reviewer. We have added the electron diffraction patterns at several different time points (–50 ps, +20 ps, +80 ps, +150 ps, and +500 ps) in the revised Fig. 4 to clarify the molecular motions in the system. The time-resolved electron diffraction setup is based on a transmission system, and the azobenzene molecules are homeotropically aligned on the substrate; therefore, the interlayer motions cannot be observed. According to the revised Fig. 4a, the degree of rotation is almost fixed just after the photoexcitation (in <50 ps), and the number of rotating domains increases with a time constant of 170 (± 10) ps. This suggests that a small number of bent azobenzene molecules creates rotating domains. The bending motion of the molecules simultaneously triggers the rotational motion of the surrounding molecules, which agrees well with the following discussion (reviewer #2, comment #4) that it takes 180–190 (± 20) ps to undergo the structural change (bending motion). The rotating domains connect to each other and grow as time evolves to result in an $\sim 10\%$ better orientation. Thanks to this reviewer's comment, we have provided a more convincing explanation in our manuscript. (page 9 blue marked, Revised Fig. 4)

2. More information is needed on the transient absorption data. I believe the transient absorption data is critical to the storyline because the appearance of the $n-\pi^$ band is the only evidence of the occurrence of the isomerization. The transient transmission data show a decay but is less convincing to conclude that this is isomerization. The author needs to provide more information on the transient absorption measurement. How is it measured? What's the pump-probe delay, fluence, pulse duration etc. Now the figure legend says "UV light 3 min" without any explanation what it means. This is really confusing because a pump-probe delay of 3 minutes is contradicting that the original phase is recovered within 5 ms after pump. In addition, considering the importance of spectroscopic studies to the storyline I suggest moving Fig S4 and S5 to the main text.*

We have performed additional spectroscopic measurement on the azobenzene sample to determine whether photoisomerization occurs or not. First, we have performed a static optical transmission experiment on *trans*-azobenzene and *cis*-azobenzene samples. According to the spectra, the transmissions at the wavelengths of 266 and 400 nm increase and decrease, respectively, by the photoisomerization. Then, we performed transient transmission experiments on the azobenzene sample. The time-evolutions of the transmission at the wavelengths of 266 and 400 nm increase and decrease upon photoexcitation at a wavelength of 266 nm. This result shows that the $n-\pi^*$ transition increases and the $\pi-\pi^*$ transition decreases under UV photoexcitation, which suggests that the *trans*-azobenzene molecules have undergone photoisomerization and bent under the UV light. We have revised the manuscript by adding all the information and the procedure of the transient transmission experiments. We have also performed the transient transmission experiment with different wavelengths (e.g., 350 nm, and 450 nm) using an optical parametric amplifier; however, the signal was weaker than the stability of the probe light, and we could not detect the signal from the photoisomerization of the azobenzene molecules. (pages 14–15 (Methods: "Optical spectroscopy of azobenzene molecules" and "Transient transmission spectroscopy"), revised

Figs. 2, S4, and S5)

3. The MD simulation needs to be better explained. The authors said the MD simulation is “2D”, does it mean only a single layer is considered? The data show a 12% change which clearly means more than one layer is affected, and therefore a single-layer model might not be sufficient. In addition, Fig. S22 to S27 has very little description text, and it’s not obvious what information the authors are trying to convey.

Regarding the first part of the comment, we have performed three-dimensional MD simulations with the simplified model. However, statistical results were not obtained from the three dimensional MD simulation because the model molecules often are buried between the layers that do not exhibit the smectic B (Hex) phase. As indicated in Goodby, J. W., Collings, P. J., Kato, T., Tschierske, C., Gleeson, H. F. & Raynes, P. (eds), Handbook of Liquid Crystals, 2nd edn (Wiley-VCH, Weinheim, Germany, 2014), the correlation among the interlayer molecules is much weaker than that among the intralayer molecules in the smectic B (Hex) phase. Therefore, we conclude that two-dimensional MD simulation is more appropriate in this work.

The other part of the comment is related to major comment #9 from reviewer #1. We have added explanations for Figs. S23–S28 in the supplementary information. The two-dimensional MD simulation was performed on azobenzene molecules at various densities (0.96, 0.97, and 0.98). The condition of the density at 1.00 corresponds to close packing. The azobenzene molecule is ellipsoidal, and the sizes of the *trans*- and *cis*-azobenzene molecules are 1:1:10 (straight form) and 1:3:8 (bent form), respectively. One hundred azobenzene molecules are contained in a unit system, and the photoexcitation can change an azobenzene molecule from the straight form to the bent form (1%). The azobenzene molecules in the LC phase before and after the photoexcitation are represented by the systems without and with a bent molecule, respectively. Figures S24, S26, and S28 show 20 calculated electron diffraction patterns (on the timescale of 5–100 LJt) at densities of 0.96, 0.97, and 0.98, respectively. As shown in Fig. S28, the six-fold symmetry of the electron diffraction patterns increases with the addition of a bent molecule at a higher density (0.98), *i.e.*, the azobenzene molecules in the liquid-crystalline phase become more oriented with the addition of bent molecules.

4. The authors concluded “the trans-to-cis photoisomerization occurs on the timescale of ~110 ps” from the transient transmission data, but it is questionable. The transient transmission data show a ~110 ps dynamics, but why is this not the same dynamics as observed in UED, which occurs at a very similar timescale? I believe the photoisomerization itself should be very fast (typically sub-picosecond) because it’s an intramolecular motion, and the subsequent large-scale re-orientation could be much slower (~100 ps), because it is induced by intermolecular interaction. The authors should be more careful when describing photoisomerization vs higher orientation.

We greatly appreciate the reviewer since this comment is quite important. The dynamics of the photoisomerization of azobenzene molecules have been well investigated in the solution phase. These reports addressed that the processes of photoisomerization can be typically separated into three parts, *i.e.*, the electronic π - π^* transition, the electronic transition to the

conical intersection, and the structural change from the *trans*-form to the *cis*-form. As reviewer noted, these studies have illustrated that the electronic π - π^* transition and the electronic transition to the conical intersection occur quite rapidly (<1 ps). The structural change (*trans*-to-*cis* isomerization) also occurs in 1–10 ps. The structural change strongly depends on the surrounding environment. Azobenzene molecules in a low-viscosity solution change their shapes within 1 ps, while azobenzene molecules in high-viscosity solution change their shapes in >10 ps. Liquid crystals are an extremely high-viscosity solution, and the azobenzene molecules in the LC phase should thus change (bend) very slowly (180–190 (± 20) ps). The change in shape of the azobenzene molecules simultaneously triggers the intermolecular motions that are observed in ultrafast time-resolved electron diffraction ($\tau = 170 \pm 10$). These timescales are in good agreement. (pages 6–7 blue marked, Fig. 2)

Above are my main concerns, there are some minor issues also need to be addressed:

We appreciate the reviewer's comments and suggestions and we have revised our manuscript accordingly.

1. The author uses "azobenzene molecule" or "azobenzene" to represent azobenzene derivative throughout the text. It is not a big problem but it's still good to clarify at some point in the text.

We have standardized the fluctuation of "azobenzene" and "azobenzene molecule".

2. The definition of Q. In the UED community it is a convention that Q represents momentum transfer ($Q=4\pi/\lambda \cdot \sin(\theta/2)$), and I believe there's a factor of 4π from the author's definition. It needs to be clarified somewhere.

We have clarified the definition of Q in the caption of the revised Fig. 3a.

3. Fig. 2a is missing color bar.

We have added the colour bars in the revised Figs. 3a.

4. The text "the molecular packing distance between the azobenzene molecules was 5.1 Å, and the azobenzene molecules should be homeotropically aligned and resides in a smectic B (hexatic) phase" needs some reference. Where was the number 5.1 Å coming from? How did the authors conclude that the azobenzene molecules are in this phase?

The packing distance of the intralayer molecules can be calculated from the electron diffraction pattern shown in the revised Fig. 3a. The distance of the diffraction spots is $Q = 4\pi \times 0.22 \text{ \AA}^{-1}$, which means that the plane distance is 4.5 Å. The plane distance and molecular distance have a relation of $1:2/\sqrt{3}$ for a hexagonal lattice (see the figure in next page), therefore, the packing distance of the molecules is calculated to be $4.5 \times 2/\sqrt{3} = 5.1 \text{ \AA}$. The six-fold symmetric diffraction pattern, as shown in the revised Fig. 3a, also suggests that the azobenzene molecules exhibit the smectic B phase. We have added a relevant reference

regarding the relation of the diffraction pattern and the liquid-crystalline phase (Goodby, J. W. et al, X-Ray Observation of a Stacked Hexatic Liquid-Crystal B Phase, Phys. Rev. Lett. 46, 1135-1138 (1981).). (page 8 blue marked)

5. Fig. 3a and 3b should have something to show the original Bragg peak direction, something like the white dashed lines in Fig. 4 will be good.

We have added black dashed lines in the revised Figs. 4a and 4b to guide the eyes.

6. Fig. S9 shows the UED for crystalline phase sample, which essentially nothing changed after time zero. However, there should at least be some heating effect (Debye-Waller), since the crystalline sample has comparable absorbance at 266 nm as the LC sample. Can the authors explain why there's no Debye Waller effect?

The Debye-Waller effect appears as the intensity decreases and this effect is more significant for higher-ordered diffraction spots or rings. As shown in the DSC measurements, this molecule exhibits a liquid-crystalline phase in a quite narrow temperature window, 50–60 °C, and it exhibits a liquid phase above 60 °C. The six-fold symmetric patterns are observed after photoexcitation, which suggests that the sample was not melted by photoexcitation and thus the sample was not heated by 10 °C by photoexcitation. The Debye-Waller effect by a change in the temperature of <10 °C is negligibly small for a first-order diffraction ring. Thus, it is natural to think that the incident UV light does not heat the sample sufficiently to detect any Debye-Waller effect in the electron diffraction pattern.

7. There are many typos in the manuscript:

-Page 5, Fig. S5 and S6 should be S4 and S5

-Page 9, there's no Fig. 3e

-Page 11, "Conventional transient absorption spectroscopy" should be "Conventional transient transmission spectroscopy".

We have corrected the typos accordingly.

In sum, I cannot support the manuscript to be published in Nature Communications, at least in its current form.

REVIEWERS' COMMENTS:

Reviewer #1 (Remarks to the Author):

The authors have collected additional data to support their conclusions and addressed the reviewers' comments. At this point, I suggest the manuscript for publication. However, due to a larger number of typos and grammatical errors in the main text and supplementary materials, I strongly recommend that those will be edited by someone with good English writing skills and some background in Physical Chemistry. After all, the investigators put a lot of effort to complete this study, so it would be a pity to reduce the impact of this work because of the language issue.

Reviewer #2 (Remarks to the Author):

In the revised manuscript, Dr. Kato and co-authors have made significant improvement in comparison to their original manuscript. The additional transient spectroscopy data complements the UED data very well and the story is now much more convincing. Explanation on both UED data and MD simulation are also consolidated. The first direct observation of isomerization in liquid crystal phase using UED will generate strong scientific interest for the relevant communities.

I recommend the acceptance of this manuscript in Nature Communications.

I thank the reviewers for their helpful comments about our manuscript. I carefully revised our manuscript according to the comments.

Comments to Reviewer #1

The authors have collected additional data to support their conclusions and addressed the reviewers' comments. At this point, I suggest the manuscript for publication. However, due to a larger number of typos and grammatical errors in the main text and supplementary materials, I strongly recommend that those will be edited by someone with good English writing skills and some background in Physical Chemistry. After all, the investigators put a lot of effort to complete this study, so it would be a pity to reduce the impact of this work because of the language issue.

I appreciate that the reviewer evaluated our manuscript as suitable for publication in *Nature Communications*. As the reviewer suggested, an expert in physical chemistry carefully checked and corrected the manuscript and supplementary information. I also asked native English speakers from an editing service (Nature Research Editing Service) to proofread the manuscript. I thoroughly checked the manuscript and corrected all the typos and other errors.

Comments to Reviewer #2

In the revised manuscript, Dr. Kato and co-authors have made significant improvement in comparison to their original manuscript. The additional transient spectroscopy data complements the UED data very well and the story is now much more convincing. Explanation on both UED data and MD simulation are also consolidated. The first direct observation of isomerization in liquid crystal phase using UED will generate strong scientific interest for the relevant communities.

I recommend the acceptance of this manuscript in Nature Communications

We are grateful that the reviewer recommended our work for acceptance in *Nature Communications*.